# Phenotypic landscape inference reveals multiple evolutionary paths to C$_4$ photosynthesis

Ben P Williams[1†], Iain G Johnston[2†], Sarah Covshoff[1], Julian M Hibberd[1]*

[1]Department of Plant Sciences, University of Cambridge, Cambridge, United Kingdom; [2]Department of Mathematics, Imperial College London, London, United Kingdom

**Abstract** C$_4$ photosynthesis has independently evolved from the ancestral C$_3$ pathway in at least 60 plant lineages, but, as with other complex traits, how it evolved is unclear. Here we show that the polyphyletic appearance of C$_4$ photosynthesis is associated with diverse and flexible evolutionary paths that group into four major trajectories. We conducted a meta-analysis of 18 lineages containing species that use C$_3$, C$_4$, or intermediate C$_3$–C$_4$ forms of photosynthesis to parameterise a 16-dimensional phenotypic landscape. We then developed and experimentally verified a novel Bayesian approach based on a hidden Markov model that predicts how the C$_4$ phenotype evolved. The alternative evolutionary histories underlying the appearance of C$_4$ photosynthesis were determined by ancestral lineage and initial phenotypic alterations unrelated to photosynthesis. We conclude that the order of C$_4$ trait acquisition is flexible and driven by non-photosynthetic drivers. This flexibility will have facilitated the convergent evolution of this complex trait.

*For correspondence: Julian.
Hibberd@plantsci.cam.ac.uk

†These authors contributed equally to this work

**Competing interests:** The authors declare that no competing interests exist.

**Reviewing editor**: Dominique Bergmann, Stanford University, United States

## Introduction

The convergent evolution of complex traits is surprisingly common, with examples including camera-like eyes of cephalopods, vertebrates, and cnidaria (*Kozmik et al., 2008*), mimicry in invertebrates and vertebrates (*Santos et al., 2003*; *Wilson et al., 2012*) and the different photosynthetic machineries of plants (*Sage et al., 2011a*). While the polyphyletic origin of simple traits (*Hill et al., 2006*; *Steiner et al., 2009*) is underpinned by flexibility in the underlying molecular mechanisms, the extent to which this applies to complex traits is less clear. C$_4$ photosynthesis is both highly complex, involving alterations to leaf anatomy, cellular ultrastructure, and photosynthetic metabolism, and also convergent, being found in at least 60 independent lineages of angiosperms (*Sage et al., 2011a*). As the emergence of the entire C$_4$ phenotype cannot be comprehensively explored experimentally, C$_4$ photosynthesis is an ideal system for the mathematical modelling of complex trait evolution as transitions on an underlying phenotype landscape. Furthermore, understanding the evolutionary events that have generated C$_4$ photosynthesis on many independent occasions has the potential to inform approaches being undertaken to engineer C$_4$ photosynthesis into C$_3$ crop species (*Hibberd et al., 2008*).

The C$_4$ pathway is estimated to have first evolved between 32 and 25 million years ago (*Christin et al., 2011b*) in response to multiple ecological drivers, including decreasing atmospheric CO$_2$ concentration (*Vicentini et al., 2008*). C$_4$ species have since radiated to represent the most productive crops and native vegetation on the planet because modifications to their leaves increase the efficiency of photosynthesis in the sub-tropics and tropics (*Edwards et al., 2010*). In C$_4$ plants, photosynthetic efficiency is improved compared with C$_3$ species because significant alterations to leaf anatomy, cell biology and biochemistry lead to higher concentrations of CO$_2$ around the primary carboxylase RuBisCO *Slack and Hatch, 1967*; *Langdale, 2011*). The morphology of C$_4$ leaves is typically modified into so-called Kranz anatomy that consists of repeating units of vein, bundle sheath (BS) and mesophyll (M) cells (*Hattersley, 1984*; *Langdale, 2011*) (*Figure 1—figure supplement 1*). Photosynthetic metabolism

**eLife digest** Plants rely on carbon for their growth and survival: in a process called photosynthesis, they use energy from sunlight to convert carbon dioxide and water into carbohydrates and oxygen gas. The chemical reactions that make up photosynthesis are powered by a chain of enzymes, and plants must ensure that these enzymes—which are in the leaves of the plant—are supplied with enough carbon dioxide and water. Carbon dioxide from the atmosphere enters plants through pores in their leaves, but water must be carried up the plant from the roots.

The type of photosynthesis used by about 90% of flowering plant species—including tomatoes and rice—is called $C_3$ photosynthesis. The first step in this process begins with an enzyme called RuBisCO, which reacts with carbon dioxide and a substance called RuBP to form molecules that contain three carbon atoms (hence the name $C_3$ photosynthesis).

In a hot climate, however, a plant can lose a lot of water through the pores in its leaves: closing these pores allows the plant to retain water, but this also reduces the supply of carbon dioxide. Under these circumstances this causes problems because RuBisCO uses oxygen to break down RuBP, instead of creating sugars, when carbon dioxide is not readily available. To prevent this process, which wastes a lot of energy and resources, some plants—including maize, sugar cane and many other agricultural staples—have evolved an alternative process called $C_4$ photosynthesis. Although it is more complex than $C_3$ photosynthesis, and required many changes to be made to the structure of leaves, $C_4$ photosynthesis has evolved on more than 60 different occasions.

In $C_4$ plants, the mesophyll—the region that is associated with the capture of carbon dioxide by RuBisCO in $C_3$ plants—contains high levels of an alternative enzyme called PEPC that converts carbon dioxide molecules into an acid that contains four carbon atoms. To avoid carbon dioxide being captured by both enzymes, $C_4$ plants evolved to relocate RuBisCO from the mesophyll to a second set of cells in an airtight structure known as the bundle sheath. The four-carbon acids produced by PEPC diffuse to the cells in the bundle sheath, where they are broken down into carbon dioxide molecules, and photosynthesis then proceeds as normal. This process allows photosynthesis to continue when the level of carbon dioxide in the leave is low because the plant has closed its pores to retain water.

Since $C_4$ plants grow faster than $C_3$ plants, and also require less water, plant biologists would like to introduce certain $C_4$ traits into $C_3$ crop plants. To help with this process, Williams, Johnston et al. have used computational methods to explore how $C_4$ photosynthesis evolved from ancestral $C_3$ plants. This involved investigating the prevalence of 16 traits that are common to $C_4$ plants in a total of 73 species that undergo $C_3$ or $C_4$ photosynthesis (including 37 species that possess characteristics of both $C_3$ and $C_4$).

Williams, Johnston et al. then went on to produce a new mathematical model that represents evolutionary processes as pathways across a multi-dimensional "landscape". The model shows that traits can be acquired in various orders, and that $C_4$ photosynthesis evolved through a number of independent pathways. Some traits that evolved early in the transitions to $C_4$ photosynthesis influenced how evolution proceeded, providing "foundations" upon which further changes evolved.

Interestingly, the structure of the leaf itself appeared to change before any of the photosynthetic enzymes changed. This led Williams, Johnston et al. to conclude that climate change—in particular, the declines in carbon dioxide levels that occurred in prehistoric times—was probably not responsible for the original evolution of $C_4$ photosynthesis. Nevertheless, these results could help with efforts to adapt important $C_3$ crop plants to on-going changes in our climate.

becomes modified and compartmentalised between the M and BS, with M cells lacking RuBisCO but instead containing high activities of the alternate carboxylase PEPC to generate $C_4$ acids. The diffusion of these acids followed by their decarboxylation in BS cells around RuBisCO increases $CO_2$ supply and therefore photosynthetic efficiency (*Zhu et al., 2008*). $C_4$ acids are decarboxylated by at least one of three enzymes within BS cells: NADP- or NAD-dependent malic enzymes (NADP-ME or NAD-ME respectively), or phospho*enol*pyruvate carboxykinase (PCK) (*Hatch et al., 1975*). Specific lineages of $C_4$ species have typically been classified into one of three sub-types, based on the activity of these decarboxylases, as well as anatomical and cellular traits that consistently correlate with each other (*Furbank, 2011*).

The genetic mechanisms underlying the evolution of cell-specific gene expression associated with the separation of photosynthetic metabolism between M and BS cells involve both alterations to *cis*-elements and *trans*-acting factors (*Akyildiz et al., 2007*; *Brown et al., 2011*; *Kajala et al., 2012*; *Williams et al., 2012*). Phylogenetically independent lineages of $C_4$ plants have co-opted homologous mechanisms to generate cell specificity (*Brown et al., 2011*) as well as the altered allosteric regulation of $C_4$ enzymes (*Christin et al., 2007*) indicating that parallel evolution underpins at least part of the convergent $C_4$ syndrome. However, while a substantial amount of work has addressed the molecular alterations that generate the biochemical differences between $C_3$ and $C_4$ plants (*Williams et al., 2012*) much less is known about the order and flexibility with which phenotypic traits important for $C_4$ photosynthesis are acquired (*Sage et al., 2012*). Clues to this question exist in the form of $C_3$–$C_4$ intermediates, species exhibiting characteristics of both $C_3$ or $C_4$ photosynthesis, such as the activity or localisation of $C_4$ cycle enzymes (*Hattersley and Stone, 1986*), the possession of one or more anatomical or cellular adaptations associated with $C_4$ photosynthesis (*Moore et al., 1987*), or combinations of both (e.g., *Kennedy et al., 1980; Kotayeva et al., 2010*). To address these unknown aspects of $C_4$ evolutionary history, we combined the concept of considering evolutionary paths as stochastic processes on complex adaptive landscapes (*Wright, 1932*; *Gavrilets, 1997*) with the analysis of extant $C_3$–$C_4$ intermediate species to develop a predictive model of how the full $C_4$ phenotype evolved.

## Results

### A meta-analysis of photosynthetic phenotypes

To parameterise the phenotypic landscape underlying photosynthetic phenotypes, data was consolidated from 43 studies encompassing 18 $C_3$, 18 $C_4$, and 37 $C_3$–$C_4$ intermediate species from 22 genera (*Table 1*). These $C_3$–$C_4$ species are from 18 independent lineages likely representing 18 distinct evolutionary origins of $C_3$–$C_4$ intermediacy (*Sage et al., 2011a*) (*Figure 1—figure supplement 2*). These studies were used to quantify 16 biochemical, anatomical, and cellular characteristics associated with $C_4$ photosynthesis (*Figure 1—source data 1*). Principal components analysis (PCA) was performed to confirm the phenotypic intermediacy of the $C_3$–$C_4$ species (*Figure 1A*). This result, the sister-group relationships of $C_3$–$C_4$ species with congeneric $C_4$ clades (*McKown et al., 2005*; *Vogan et al., 2007*; *Christin et al., 2011a*; *Sage et al., 2011a*; *Khoshravesh et al., 2012*) and the prevalence of extant $C_3$–$C_4$ species in genera with the most recent origins of $C_4$ photosynthesis (*Christin et al., 2011b*) all support the notion that $C_3$–$C_4$ species represent phenotypic states through which transitions to $C_4$ photosynthesis could occur. The combined traits of $C_3$–$C_4$ intermediate species therefore represent samples from across the space of phenotypes connecting $C_3$ to $C_4$ photosynthesis (*Figure 1B*). Within our meta-analysis data, $C_3$–$C_4$ phenotypes were available for 33 eudicot and 4 monocot species. 16 and 17 of these species have extant congeneric relatives performing NADP-ME or NAD-ME subtype $C_4$ photosynthesis respectively. No $C_3$–$C_4$ relatives of PCK sub-type $C_4$ species are known (*Sage et al., 2011a*). Our meta-analysis therefore encompassed a variety of taxonomic lineages, as well as representing close relatives of known phenotypic variants performing $C_4$ photosynthesis.

We defined each $C_4$ trait as either being absent (0) or present (1). For quantitative traits the expectation-maximization (EM) algorithm and hierarchical clustering were used to impartially assign binary scores (*Figure 1—figure supplement 3*). This generated a 16-bit string for each of the species (*Figure 1—source data 1*), with a presence or absence score for each of the traits included in our meta-analysis. This defined a 16-dimensional phenotype space with $2^{16}$ (65,536) nodes corresponding to all possible combinations of presence (1) and absence (0) scores for each characteristic.

### A novel Bayesian approach for predicting evolutionary trajectories

Many existing methods of inference for evolutionary trajectories rely on phylogenetic information or assumptions about the fitness landscape underlying evolutionary dynamics (*Weinreich et al., 2005*; *Lobkovsky et al., 2011*; *Mooers and Heard, 2013*). In convergent evolution, these properties are not always known, as convergent lineages may be genetically distant and associated with poor phylogenetic reconstructions. In addition, the selective pressures experienced by each may be different and dynamic. We therefore consider the convergent evolution of $C_4$ fundamentally as the acquisition of the key phenotypic traits identified through our meta-analysis (*Figure 1B*). The process of acquisition of these traits can be pictured as a path on the 16-dimensional hypercube (*Figure 1C*), from the node labelled with all 0's (the $C_3$ phenotype, with no $C_4$ characteristics) to the node labelled with all 1's (the $C_4$ phenotype, with all $C_4$ characteristics).

**Table 1.** Summary of $C_3$–$C_4$ lineages assessed

| Family | Species | References* |
|---|---|---|
| Amaranthaceae | *Alternanthera ficoides* ($C_3$–$C_4$) | *Rajendrudu et al. (1986)* |
| | *Alternanthera tenella* ($C_3$–$C_4$) | *Devi and Raghavendra (1993)* |
| | *Alternanthera pungens* ($C_4$) | *Devi et al. (1995)* |
| Asteraceae | *Flaveria cronquistii* ($C_3$) | |
| | *Flavera pringlei* ($C_3$) | |
| | *Flaveria robusta* ($C_3$) | |
| | *Flaveria angustifolia* ($C_3$–$C_4$) | |
| | *Flaveria anomala* ($C_3$–$C_4$) | *Ku et al. (1983)* |
| | *Flaveria chloraefolia* ($C_3$–$C_4$) | *Holaday et al. (1984)* |
| | *Flaveria floridana* ($C_3$–$C_4$) | *Adams et al. (1986)* |
| | *Flaveria linearis* ($C_3$–$C_4$) | *Brown and Hattersley (1989)* |
| | *Flaveria oppositifolia* ($C_3$–$C_4$) | *Ku et al. (1991)* |
| | *Flaveria ramosissima* ($C_3$–$C_4$) | *Rosche et al. (1994)* |
| | *Flaveria sonorensis* ($C_3$–$C_4$) | *Casati et al. (1999)* |
| | *Flaveria brownie* ($C_3$–$C_4$) | *McKown et al. (2005)* |
| | *Flaveria vaginata* ($C_3$–$C_4$) | *McKown and Dengler (2007)* |
| | *Flaveria pubescens* ($C_3$–$C_4$) | *Gowik et al. (2011)* |
| | *Flaveria australasica* ($C_4$) | |
| | *Flaveria bidentis* ($C_4$) | |
| | *Flaveria kochiana* ($C_4$) | |
| | *Flaveria trinervia* ($C_4$) | |
| | *Parthenium incanum* ($C_3$) | *Moore et al. (1987)* |
| | *Parthenium hysterophorus* ($C_3$–$C_4$) | *Devi and Raghavendra (1993)* |
| Boraginaceae | *Heliotropium europaeum* ($C_3$) | |
| | *Heliotropium calcicola* ($C_3$) | *Vogan et al. (2007)* |
| | *Heliotropium convolvulaceum* ($C_3$–$C_4$) | *Muhaidat et al. (2011)* |
| | *Heliotropium greggii* ($C_3$–$C_4$) | |
| | *Heliotropium polyphyllum* ($C_4$) | |
| Brassicaceae | *Moricandia foetida* ($C_3$) | *Holaday et al. (1981)* |
| | *Moricandia arvensis* ($C_3$–$C_4$) | *Rawsthorne et al. (1988)* |
| | *Moricandia spinosa* ($C_3$–$C_4$) | *Beebe and Evert (1990)* |
| | *Moricandia nitens* ($C_3$–$C_4$) | *Rawsthorne et al. (1998)* |
| | *Raphanus sativus* ($C_3$) | *Ueno et al. (2003)* |
| | *Diplotaxis muralis* ($C_3$–$C_4$) | *Ueno et al. (2006)* |
| | *Diplotaxis tenuifolia* ($C_3$–$C_4$) | |
| Chenopodiaceae | *Salsola oreophila* ($C_3$) | *P'yankov et al. (1997)* |
| | *Salsola arbusculiformis* ($C_3$–$C_4$) | *Voznesenskaya et al. (2001)* |
| | *Salsola arbuscula* ($C_4$) | |
| Cleomaceae | *Cleome spinosa* ($C_3$) | *Voznesenskaya et al. (2007)* |
| | *Cleome paradoxa* ($C_3$–$C_4$) | *Koteyeva et al. (2010)* |
| | *Cleome gynandra* ($C_4$) | |
| Cyperaceae | *Eleocharis acuta* ($C_3$) | *Bruhl and Perry (1995)* |
| | *Eleocharis acicularis* ($C_3$–$C_4$) | *Keeley (1999)* |
| | *Eleocharis tetragona* ($C_4$) | |

*Table 1. Continued on next page*

*Table 1. Continued*

| Family | Species | References* |
|---|---|---|
| Euphorbiaceae | *Euphorbia angusta* ($C_3$) | |
| | *Euphorbia acuta* ($C_3$–$C_4$) | *Sage et al. (2011b)* |
| | *Euphorbia lata* ($C_3$–$C_4$) | |
| | *Euphorbia mesembryanthemifolia* ($C_4$) | |
| Molluginaceae | *Mollugo tenella* ($C_3$) | |
| | *Mollugo verticillata* ($C_3$–$C_4$) | *Sayre et al. (1979)* |
| | *Mollugo naudicalis* ($C_3$–$C_4$) | *Kennedy et al. (1980)* |
| | *Mollugo pentaphylla* ($C_3$–$C_4$) | *Christin et al. (2011a)* |
| | *Mollugo cerviana* ($C_4$) | |
| Poaceae | *Avena sativa* ($C_3$) | *Slack and Hatch (1967)* |
| | *Neurachne tenuifolia* ($C_3$) | *Hattersley and Stone (1986)* |
| | *Neurachne minor* ($C_3$–$C_4$) | *Brown and Hattersley (1989)* |
| | *Neurachne munroi* ($C_4$) | |
| | *Panicum bisculatum* ($C_3$) | *Goldstein et al. (1976)* |
| | *Panicum hians* ($C_3$–$C_4$) | *Ku et al. (1976)* |
| | *Panicum milioides* ($C_3$–$C_4$) | *Ku and Edwards (1978)* |
| | *Panicum miliaceum* ($C_4$) | *Rathnam and Chollet (1978)* |
| | | *Rathnam and Chollet (1979)* |
| | | *Holaday and Black (1981)* |
| | | *Hattersley (1984)* |
| | *Saccharum officinarum* ($C_4$) | *Slack and Hatch (1967)* |
| | *Sorghum bicolor* ($C_4$) | *Slack and Hatch (1967)* |
| | *Triticum aestivum* ($C_3$) | *Slack and Hatch (1967)* |
| | *Zea mays* ($C_4$) | *Slack and Hatch (1967)* |
| Portulaceae | *Sesuvium portulacastrum* ($C_3$) | |
| | *Portulaca cryptopetala* ($C_3$–$C_4$) | *Voznesenskaya et al. (2010)* |
| | *Portulaca oleracea* ($C_4$) | |
| Scrophularaceae | *Anticharis kaokoensis* ($C_3$) | *Khoshravesh et al. (2012)* |
| | *Anticharis ebracteata* ($C_3$–$C_4$) | |
| | *Anticharis imbricate* ($C_3$–$C_4$) | |
| | *Anticharis namibensis* ($C_3$–$C_4$) | |
| | *Anticharis glandulosa* ($C_4$) | |

The family, species, photosynthetic type and original study are listed. In total, 16 characteristics relating to $C_4$ photosynthesis were extracted from 43 studies encompassing 18 $C_3$, 18 $C_4$, and 37 $C_3$–$C_4$ intermediate species. *References apply to all species within each genus.

The phenotypic landscape underlying the evolution of $C_4$ photosynthesis was then modelled as a transition network, with weighted edges describing the probability of transitions occurring between two phenotypic states (two nodes on the hypercube, *Figure 1—figure supplement 4*). Observed intermediate points were then used to constrain the structure of these phenotypic landscapes. To do this, we developed inferential machinery based on the framework of Hidden Markov Models (HMMs) (*Rabiner, 1989*) (*Figure 1—figure supplement 4*) and simulated an ensemble of Markov chains on trial transition networks. Each of these chains represents a possible evolutionary pathway from $C_3$ to $C_4$, and passes through several intermediate phenotypic states. The likelihood of observing intermediate states with characteristics compatible with the biologically observed data on $C_3$–$C_4$ intermediates was recorded for the set of paths supported on each trial network. A Bayesian MCMC procedure was used

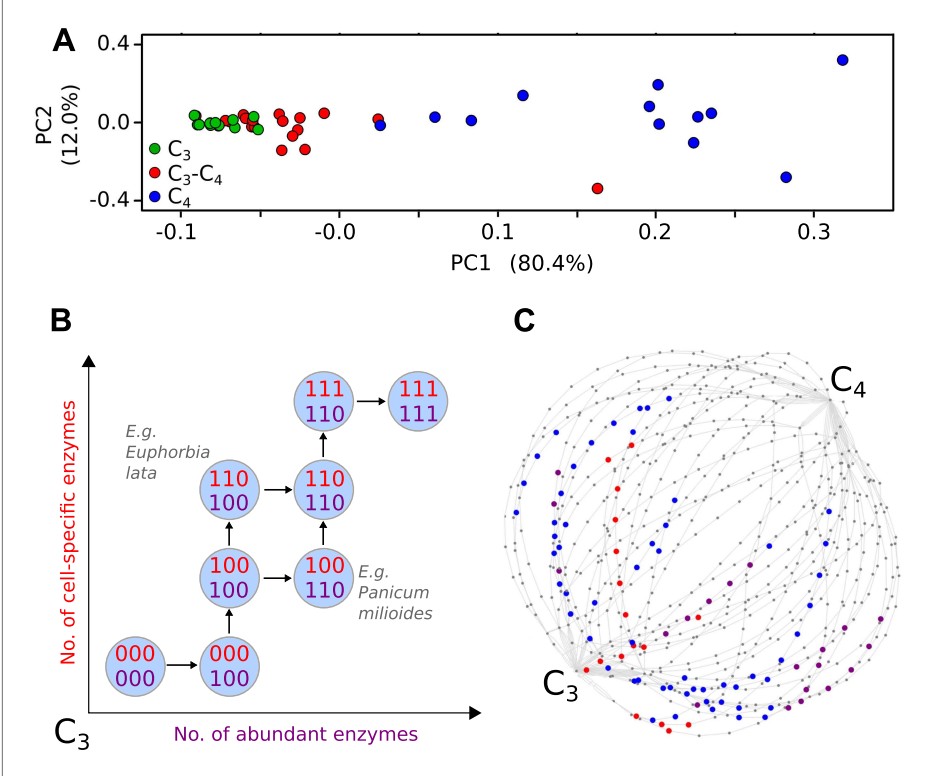

**Figure 1**. Evolutionary paths to $C_4$ phenotype space modelled from a meta-analysis of $C_3$–$C_4$ phenotypes. Principal component analysis (PCA) on data for the activity of five $C_4$ cycle enzymes confirms the intermediacy of $C_3$–$C_4$ species between $C_3$ and $C_4$ phenotype spaces (**A**). Each $C_4$ trait was considered absent in $C_3$ species and present in $C_4$ species, with previously studied $C_3$–$C_4$ intermediate species representing samples from across the phenotype space (**B**). With a dataset of 16 phenotypic traits, a 16-dimensional space was defined. (**C**) A 2D representation of 50 pathways across this space. The phenotypes of multiple $C_3$–$C_4$ species were used to identify pathways compatible with individual species (e.g., *Alternanthera ficoides* [red nodes] and *Parthenium hysterophorus* [blue nodes]), and pathways compatible with the phenotypes of multiple species (purple nodes).

The following source data and figure supplements are available for figure 1:

**Source data 1**. Binary scoring of $C_4$ traits present in $C_3$–$C_4$ species.

**Figure supplement 1**. A graphical representation of key phenotypic changes distinguishing $C_3$ and $C_4$ leaves.

**Figure supplement 2**. Phylogenetic distribution of $C_4$ and $C_3$–$C_4$ lineages across the angiosperm phylogeny.

**Figure supplement 3**. Clustering quantitative traits by EM algorithm and hierarchical clustering.

**Figure supplement 4**. Illustration of the principle by which evolutionary pathways emit intermediate signals.

to sample from the set of networks most compatible with the meta-analysis dataset, and thus most likely to represent the underlying dynamics of $C_4$ evolution. The order in which phenotypic characteristics were acquired was recorded for paths on each network compatible with the $C_3$–$C_4$ species data, and posterior probability distributions (given uninformative priors) for the time-ordered acquisition of each $C_4$ trait were generated. For further information and mathematical details, see 'Methods'.

To model the evolutionary paths generating $C_4$ without requiring additional dimensionality, we imposed that only one $C_4$ trait may be acquired at a time, and loss of acquired $C_4$ traits was forbidden. To test if we were nevertheless able to detect traits acquired simultaneously in evolution, we tested our approach on artificial positive control datasets containing intermediate nodes representing a stepwise evolutionary sequence of events (***Figure 2A***) and an evolutionary pathway in which four traits are

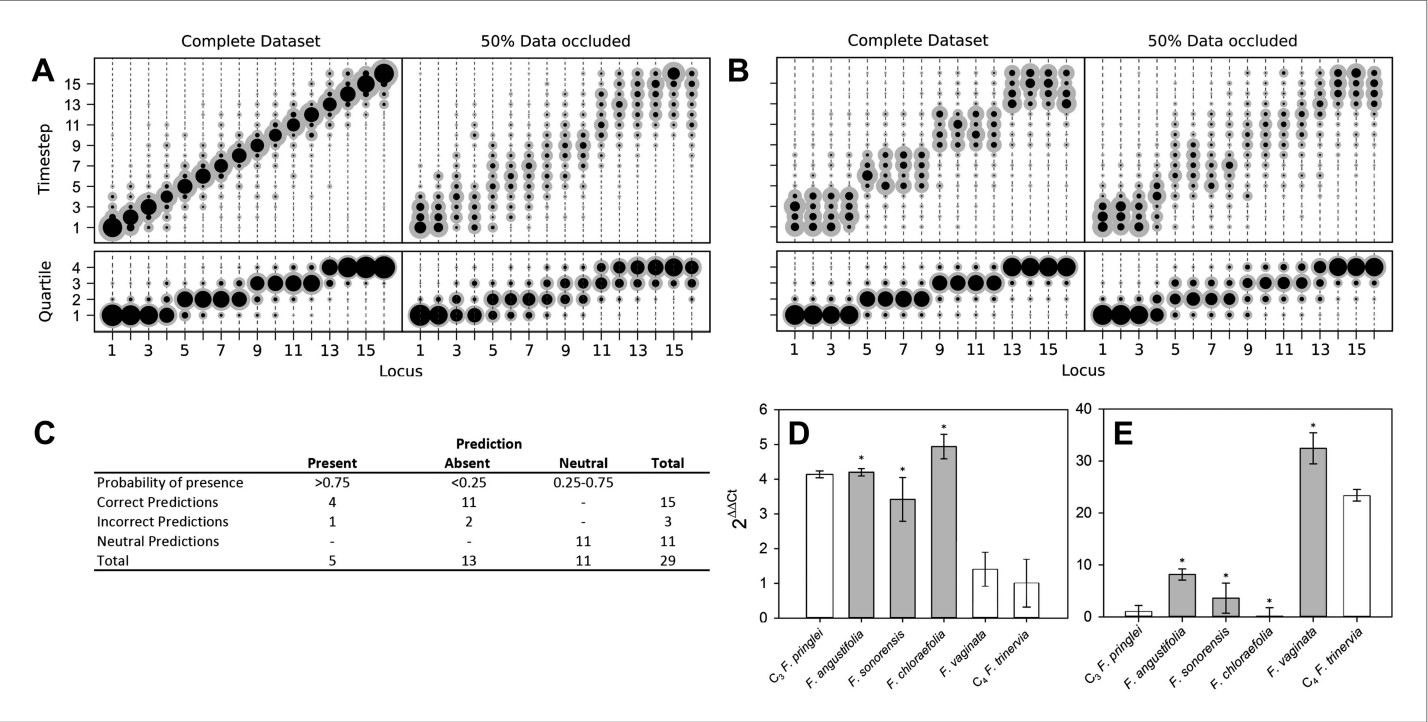

**Figure 2**. Verifying a novel Bayesian approach for predicting evolutionary trajectories. (**A** and **B**) Datasets were obtained from an artificially constructed diagonal dynamic matrix (**A**), and a diagonal matrix with linked timing of locus acquisitions (**B**). The single, diagonal evolutionary trajectory was clearly replicated in both examples, over a time-scale of 16 individual steps, or four coarse-grained quartiles. We subjected these artificial datasets to our inferential machinery with fully characterised artificial species, and with 50% of data occluded in order to replicate the proportion of missing data from our $C_3$–$C_4$ dataset. (**C**) When applied to our meta-analysis of $C_3$–$C_4$ data, predictions were generated for every trait missing from the biological dataset. We tested this predictive machinery by generating 29 artificial datasets, each missing one data point, and comparing the presence/absence of the trait as predicted by our approach with the experimental data from the original study. (**D** and **E**) Quantitative real-time PCR (qPCR) was used to verify the predicted phenotypes of four $C_3$–$C_4$ species. The abundance *RbcS* (**D**) and *MDH* (**E**) transcripts were determined from six *Flaveria* species. White bars represent phenotypes already determined by other studies, grey bars those that were predicted by the model and asterisks denote intermediate species phenotypes correctly predicted by our approach (Error bars indicate SEM, N = 3).

The following figure supplements are available for figure 2:

**Figure supplement 1**. Computational prediction of $C_3$–$C_4$ intermediate phenotypes.

acquired simultaneously at a time (***Figure 2B***). Our approach clearly assigned equal acquisition probabilities to traits whose timing was linked in the underlying dataset, even when 50% of the data was occluded (***Figure 2B***). These data are consistent with this approach detecting the simultaneous acquisition of traits in evolution, even though single-trait acquisitions are simulated.

## Verifying prediction accuracy

The presence and absence of unknown phenotypes were predicted by recording all phenotypes encountered along a set of simulated evolutionary trajectories that were compatible with the data from a given species (***Figure 1—figure supplement 4***), and calculating the posterior distribution of the proportion of these phenotypes with the value 1 for the unknown trait. If the mean of this distribution was <25% or >75%, and that value fell outside one standard deviation of the mean, the missing trait was assigned a strong prediction of absence or presence. To comprehensively test the accuracy of our predictive machinery, we generated 29 occluded datasets, consisting of the original full dataset with one randomly chosen data point removed. The predicted phenotype of each missing trait was then compared with the known phenotype published in the original study. For 29 occluded traits 18 were strongly predicted to be present or absent, and the remaining 11 predictions were neutral. Of the 18 strongly predicted traits (i.e., <25% or >75% probability), 15 were correct, with only one false positive and two false negative predictions (***Figure 2C***). The approach therefore assigns neutral predictions much more frequently than

false positive or false negative predictions, suggesting that its outputs are highly conservative, and thus unlikely to produce artefacts. Predictions were generated for phenotypes that have not yet been described in $C_3$–$C_4$ species (*Figure 2—figure supplement 1*). Quantitative real-time PCR experimentally verified a subset of these, relating to abundance of $C_4$ enzymes not previously measured (*Figure 2D–E*). We also found that the model was able to successfully infer evolutionary dynamics in artificially constructed datasets (*Figure 2A–B*). Taken together, these prediction and verification studies illustrate that our approach robustly identifies key features of $C_4$ evolution.

## A high-resolution model for the evolutionary events generating $C_4$

The posterior probability distributions for the acquisition time of each phenotypic trait were combined to produce an objective, computationally generated blueprint for the order of evolutionary events generating $C_4$ photosynthesis (*Figure 3*). These results were consistent with previous work on subsets of $C_4$ lineages that proposed the BS-specificity of GDC occurs prior to the evolution of $C_4$ metabolism (*Hylton et al., 1988*; *Rawsthorne et al., 1988*; *Devi et al., 1995*; *Sage et al., 2012*),

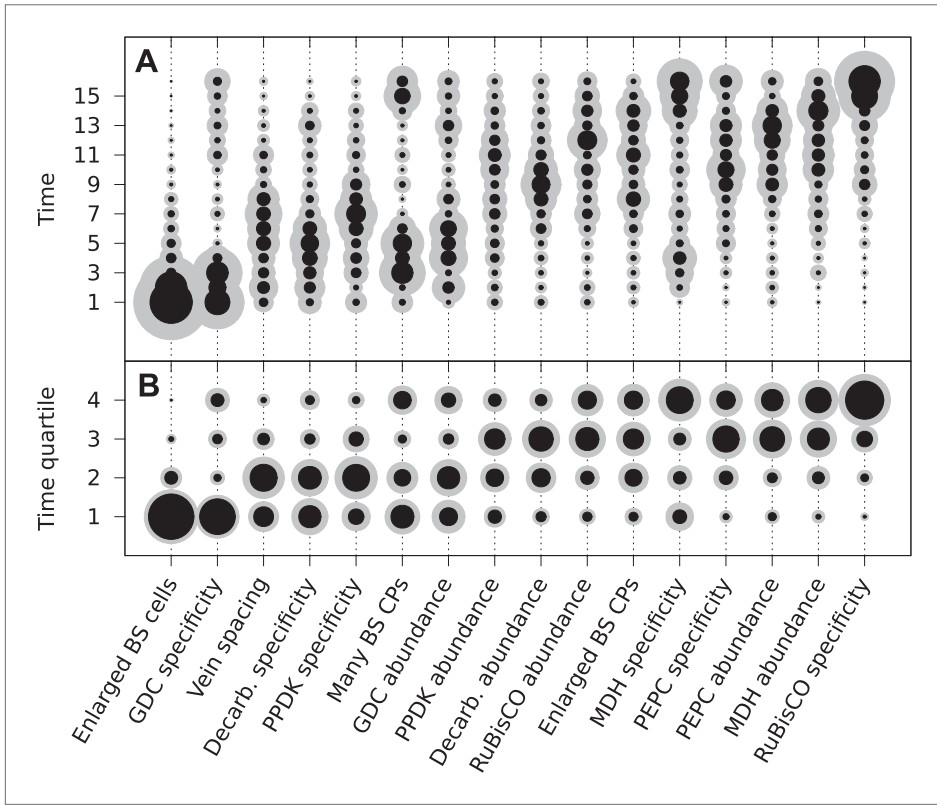

**Figure 3**. The mean ordering of phenotypic changes generating $C_4$ photosynthesis. EM-clustered data from $C_3$–$C_4$ intermediate species were used to generate posterior probability distributions for the timing of the acquisition of $C_4$ traits in sixteen evolutionary steps (**A**) or four quartiles (**B**). Circle diameter denotes the mean posterior probability of a trait being acquired at each step in $C_4$ evolution (the Bayes estimator for the acquisition probability). Halos denote the standard deviation of the posterior. The 16 traits are ordered from left to right by their probability of being acquired early to late in $C_4$ evolution. Abbreviations: bundle sheath (BS), glycine decarboxylase (GDC), chloroplasts (CPs), decarboxylase (Decarb.), pyruvate, orthophosphate dikinase (PPDK), malate dehydrogenase (MDH), phosphoenolpyruvate carboxylase (PEPC).

The following figure supplements are available for figure 3:

**Figure supplement 1**. Results obtained using data clustered by hierarchical clustering.

**Figure supplement 2**. Adding or removing traits does not affect the predicted order of evolutionary events.

**Figure supplement 3**. Probabilities of $C_4$ traits being acquired simultaneously.

and loss of RuBisCO from M cells occurs late (*Cheng et al., 1988*; *Khoshravesh et al., 2012*), but also provided higher resolution insight into the order of events generating C$_4$ metabolism. Alterations to leaf anatomy as well as cell-specificity and increased abundance of multiple C$_4$ cycle enzymes were predicted to evolve prior to any alteration to the primary C$_3$ and C$_4$ photosynthetic enzymes RuBisCO and phospho*enol*pyruvate carboxylase (PEPC) (*Figure 3*).

There was also strong evidence for enlargement of BS cells as an early innovation in most C$_4$ lineages (*Figure 3*), consistent with the suggestion that this was an ancestral state within C$_3$ ancestors of C$_4$ grass lineages and that this contributed to the high number of C$_4$ origins within this family (*Christin et al., 2013*; *Griffiths et al., 2013*). The compartmentation of PEPC into M cells and its increased abundance compared with C$_3$ leaves was predicted to occur at similar times, but for all other C$_4$ enzymes the evolution of increased abundance and cellular compartmentation were clearly separated by the acquisition of other traits (*Figure 3*). This result is consistent with molecular analysis of genes encoding C$_4$ enzymes that indicates cell-specificity and increased expression are mediated by different *cis*-elements (*Akyildiz et al., 2007*; *Kajala et al., 2012*; *Wiludda et al., 2012*).

Two approaches were taken to verify that these conclusions are robust and accurately reflect biological data. First, the analysis was repeated using scores for presence or absence of traits that were assigned by hierarchical clustering, as opposed to using the EM algorithm (*Figure 3—figure supplement 1A*). Although hierarchical clustering generated differences in the scoring of a small number of traits, the predicted evolutionary trajectories were not affected, producing highly similar results (*Figure 3—figure supplement 1B*). Second, we introduced structural changes to the phenotype space, by both adding and subtracting traits from the analysis (*Figure 3—figure supplement 2*). Removing two independent pairs of traits from the analysis did not affect the predicted timing of the remaining 14 traits (*Figure 3—figure supplement 2A–B*). However, increased standard deviations were observed in some cases (e.g., for the probabilities of acquiring enlarged BS cells, or decreased vein spacing) likely a consequence of using fewer data. To test if the addition of data might also affect the results, we performed an analysis with two additional traits included (*Figure 3—figure supplement 2C*). We selected two traits that have been widely observed in C$_3$–C$_4$ species, the centripetal positioning of mitochondria and the centrifugal or centripetal position of chloroplasts within BS cells (*Sage et al., 2012*). Despite the widespread occurrence of these traits, their functional importance remains unclear (*Sage et al., 2012*). Consistent with observations made from several genera, we predict that these cellular alterations are acquired early in the evolution of C$_4$ photosynthesis (*Hylton et al., 1988*; *McKown and Dengler, 2007*; *Muhaidat et al., 2011*; *Sage et al., 2011b*). Importantly, including these additional early traits in the analysis did not alter the predicted order of the original 16 traits. Together, these analyses did not alter our main conclusions, suggesting that they are robust.

## The order of C$_4$ trait evolution is flexible

In addition to the likely order of evolutionary events generating C$_4$ photosynthesis, the number of molecular alterations required is also unknown. We therefore aimed to test if multiple traits were predicted to evolve with linked timing, and therefore likely mediated by a single underlying mechanism. To achieve this, we performed a contingency analysis by considering trajectories across phenotype space beginning with a given initial acquisition step. In this analysis, the starting genome had one of the 16 traits acquired and the rest absent, and the contingency of the subsequent trajectory upon the initial step was recorded. This approach was designed to test if acquiring one C$_4$ trait increased the probability of subsequently acquiring other traits, thus detecting if the evolution of multiple traits is linked by underlying mechanisms. Inflexible linkage between multiple traits was detected in artificial positive control datasets (*Figure 2B*) but not in the C$_3$–C$_4$ dataset (*Figure 3—figure supplement 3*). This result suggests that the order of C$_4$ trait acquisition is flexible. Multiple origins of C$_4$ may therefore have been facilitated by this flexibility in the evolutionary pathways connecting C$_3$ and C$_4$ phenotypes.

## C$_4$ evolved via multiple distinct evolutionary trajectories

Our Bayesian analysis strongly indicates that there are multiple evolutionary pathways by which C$_4$ traits are acquired by all lineages of C$_4$ plants. First, no single sequence of acquisitions was capable of producing intermediate phenotypes compatible with all observations ('Methods'). Second, several traits such as compartmentation of GDC into BS and the increased number of chloroplasts in the BS clearly displayed bimodal probability distributions for their acquisition (*Figure 3*). This bimodality is indicative of multiple distinct pathways to C$_4$ photosynthesis that acquire traits at earlier or later times. To investigate factors

underlying this bimodality, we inferred evolutionary pathways generating the C$_4$ leaf using data from monocot and eudicot lineages, or from lineages using NAD malic enzyme (NAD–ME) or NADP malic enzyme (NADP-ME) as their primary C$_4$ acid decarboxylase. PCA on the entire set of inferred transition networks for monocot and dicot subsets revealed distinct separation (*Figure 4A*), suggesting that the topology of the evolutionary landscape surrounding C$_4$ is largely different for these two anciently diverged taxa. Performing this PCA including networks that were inferred from the full data set (with both lineages) confirmed that this separation is a robust result and involves posterior variation on a

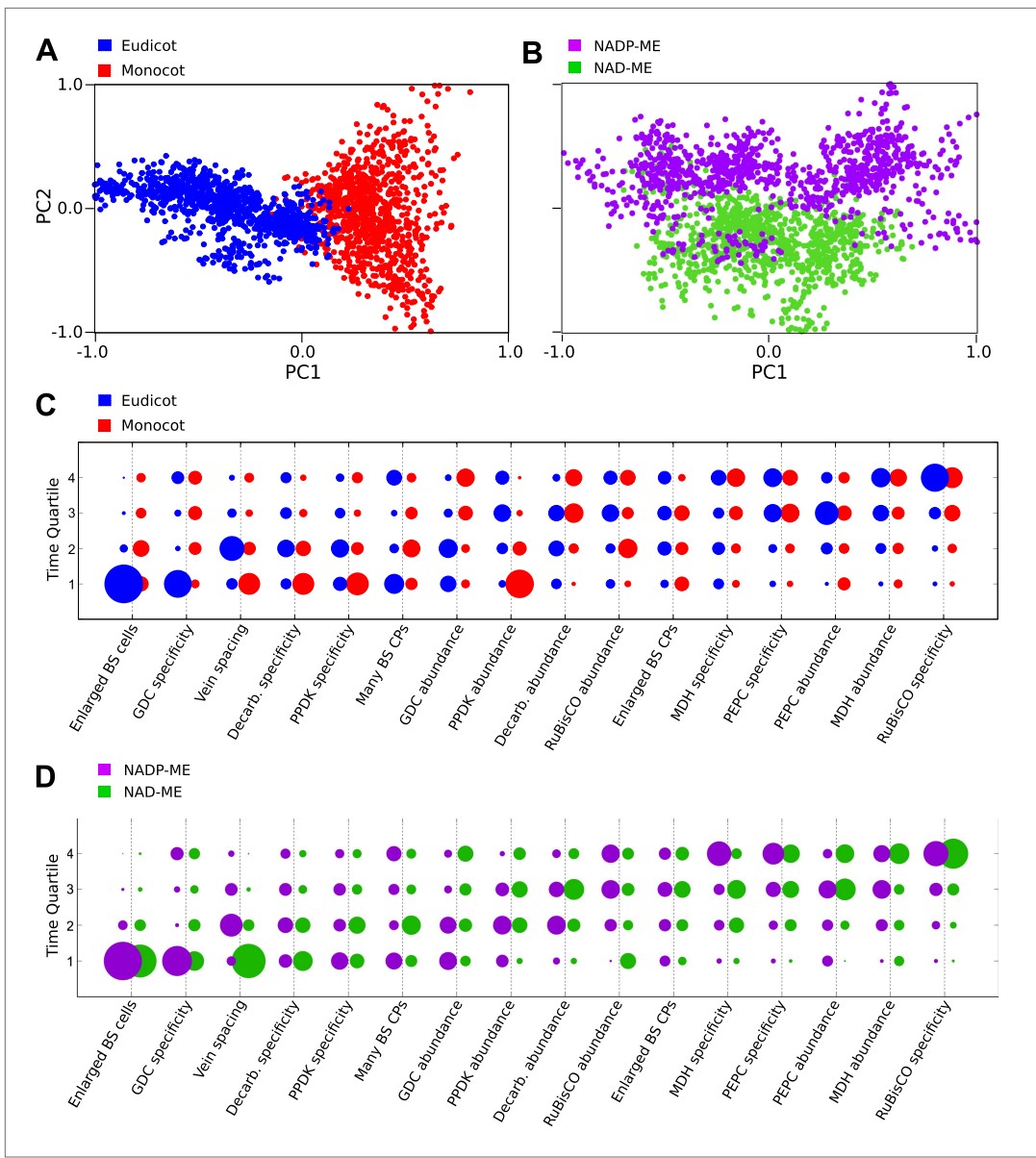

**Figure 4**. Differences in the evolutionary events generating different C$_4$ sub-types and distantly related taxa. Principal component analysis (PCA) on the entire landscape of transition probabilities using only monocot and eudicot data (**A**) and data from NADP-ME and NAD-ME sub-type lineages (**B**) shows broad differences between the evolutionary pathways generating C$_4$ in each taxon. Monocots and eudicots differ in the predicted timing of events generating C$_4$ anatomy and biochemistry (**C**), whereas NADP-ME and NAD-ME lineages differ primarily in the evolution of decreased vein spacing and greater numbers of chloroplasts in BS cells (**D**).

The following figure supplements are available for figure 4:

**Figure supplement 1**. Variation between lineages compared to variance of overall dataset.

comparable scale to that of the full set of possible networks (*Figure 4—figure supplement 1*). Analysis of the posterior probabilities of the mean pathways representing either monocots or dicots revealed that this separation is the result of differences in the timing of events generating both anatomical and biochemical traits (*Figure 4C*). We propose that the ancient divergence of the monocot and eudicot clades constrained the evolution of $C_4$ photosynthesis to broadly different evolutionary pathways in each.

There was more overlap between the landscapes generating NAD–ME and NADP-ME species (*Figure 4B*), likely reflecting the convergent origins of NAD–ME and NADP-ME sub-types (*Furbank, 2011*; *Sage et al., 2011a*). Despite the traditional definition of these lineages on the basis of biochemical differences, we detected differences in the timing of their anatomical evolution (*Figure 4D*). For example, in NAD–ME lineages, increased vein density was predicted to be acquired early in $C_4$ evolution, while in NADP-ME species this trait showed a broadly different trajectory (*Figure 4D*). The proliferation of chloroplasts in the BS was also acquired with different timings between the two sub-types. The alternative evolutionary pathways generating the NADP-ME and NAD–ME subtypes were therefore defined by differences in the timing of anatomical and cellular traits that are predicted to precede the majority of biochemical alterations (*Figure 3*, *Figure 4D*). We therefore conclude that these distinct sub-types evolved as a consequence of alternative evolutionary histories in response to non-photosynthetic pressures. Furthermore, we propose that early evolutionary events determined the downstream phenotypes of $C_4$ sub-types by restricting lineages to independent pathways across phenotype space.

## Discussion

### A novel Bayesian technique for inferring stochastic trajectories

The adaptive landscape metaphor has provided a powerful conceptual framework within which evolutionary transitions can be modelled (*Gavrilets, 1997*; *Whibley et al., 2006*; *Lobkovsky et al., 2011*). However, the majority of complex biological traits provide numerous challenges in utilising such an approach, including missing phenotypic data, incomplete phylogenetic information and in the case of convergent evolution, variable ancestral states. Here we report the development of a novel, predictive Bayesian approach that is able to infer likely evolutionary trajectories connecting phenotypes from sparsely sampled, highly stochastic data. With this model, we provided insights into the evolution of one of the most complex traits to have arisen in multiple lineages: $C_4$ photosynthesis. However, as our approach is not dependent on detailed phylogenetic inference, we propose that it could be used to model the evolution of other complex traits, such as those in the fossil record, which are also currently limited by the fragmented nature of data available (*Kidwell and Holland, 2002*). Our approach is also not limited by the time-scale over which predicted trajectories occur. As a result, it may be useful in inferring pathways underlying stochastic processes occurring over much shorter timescales, such as disease or tumour progression, or the differentiation of cell types.

### $C_4$ evolution was initiated by non-photosynthetic drivers

A central hypothesis for the ecological drivers of $C_4$ evolution is that declining $CO_2$ concentration in the Oligocene decreased the rate of carboxylation by RuBisCO, creating a strong pressure to evolve alternative photosynthetic strategies (*Christin et al., 2008*; *Vicentini et al., 2008*). According to this hypothesis, alterations to the localisation and abundance of the primary carboxylases PEPC and RuBisCO would be expected to occur early in the evolutionary trajectories generating $C_4$. Conversely, our data suggest that alterations to anatomy and cell biology were predicted to precede the majority of biochemical alterations, and that other enzymes of the $C_4$ pathway are recruited prior to PEPC and RuBisCO (*Figure 3*). These enzymes, such as PPDK and $C_4$ acid decarboxylases, function in processes not related to photosynthesis within leaves of $C_3$ plants (*Aubry et al., 2011*), so the early changes to abundance and localisation of these enzymes within $C_4$ lineages may have been driven by non-photosynthetic pressures. A recent in silico study also predicts that changes to photorespiratory metabolism and GDC in BS cells evolved prior to the $C_4$ pathway (*Heckman et al., 2013*). Our model predicts that BS-specificity of GDC was acquired early in $C_4$ evolution for the majority of lineages. However, we also note that the predicted timing of GDC BS-specificity is bimodal in our analysis (*Figure 3*), and not predicted to be acquired early in monocot lineages (*Figure 4C*). These results suggest that this is not a feature of $C_4$ evolution to have occurred repeatedly in all lineages.

Recent evidence from physiological and ecological studies has identified a number of additional environmental pressures that may have driven the evolution and radiation of $C_4$ lineages, including

high evaporative demands (*Osborne and Sack, 2012*) and increased fire frequency (*Edwards et al., 2010*). Increased BS volume and vein density have been proposed as likely adaptations to improve leaf hydraulics under drought (*Osborne and Sack, 2012*; *Griffiths et al. 2013*), but nothing is known about how early recruitment of GDC, PPDK, and $C_4$ acid decarboxylases (*Figure 3*) may relate to these pressures. A better understanding of the mechanisms underlying the recruitment of these enzymes (*Brown et al., 2011*; *Kajala et al., 2012*; *Wiludda et al., 2012*) may help identify the key molecular events facilitating $C_4$ evolution.

Our data also suggest that modifications to leaf development drove the evolution of diverse $C_4$ sub-types. For example, we find that differences in the timing of events altering leaf vascular development and BS chloroplast division occur prior to the appearance of the alternative evolutionary pathways generating the NADP-ME and NAD-ME biochemical sub-types (*Figure 4D*). These traits are predicted to evolve prior to any alterations to the $C_4$ acid decarboxylase enzymes that traditionally define these sub-types (*Furbank, 2011*). As an homologous mechanism has been shown to regulate the cell-specificity of gene expression in both NADP-ME and NAD-ME gene families in independent lineages (*Brown et al., 2011*), it is unlikely that mechanisms underlying the recruitment of these enzymes drove the evolution of distinct sub-types. We therefore conclude that these different sub-types evolved as a consequence of alternative evolutionary histories in leaf development, rather than biochemical or photosynthetic pressures. This may explain why differences in the carboxylation efficiency or photosynthetic performance of different $C_4$ sub-types have never been detected (*Furbank, 2011*), making the adaptive significance of different decarboxylation mechanisms difficult to explain. Instead, we propose that early evolutionary events determined the downstream phenotypes of $C_4$ sub-types by restricting lineages to independent pathways across phenotype space. The numerous differences in leaf development and cell biology between $C_4$ sub-types (*Furbank, 2011*) may provide clues as to which developmental changes underlie subsequent differences in metabolic evolution.

## Convergent evolution was facilitated by flexibility in evolutionary trajectories

$C_4$ photosynthesis provides an excellent example of how independent lineages with a wide range of ancestral phenotypes can converge upon similar complex traits. Several studies on more simple traits have demonstrated that convergence upon a phenotype can be specified by diverse genotypes, and thus non-homologous molecular mechanisms in independent lineages (*Wittkopp et al., 2003*; *Hill et al., 2006*; *Steiner et al., 2009*). Taken together, our data also indicate that flexibility in the viable series of evolutionary events has also facilitated the convergence of this highly complex trait. First, we show that at least four distinct evolutionary trajectories underlie the evolution of $C_4$ lineages (*Figure 4*). Second, we find no evidence for inflexible linkage between the predicted timing of distinct $C_4$ traits (*Figure 3—figure supplement 1*). This diversity in viable pathways also helps explain why $C_4$ has been accessible to such a wide variety of species and not limited to a smaller subset of the angiosperm phylogeny. A recent model for the evolution of the biochemistry associated with the $C_4$ leaf also found that $C_4$ photosynthesis was accessible from any surrounding point of a fitness landscape (*Heckman et al., 2013*). Our study of $C_4$ anatomy, biochemistry, and cell biology also suggests the $C_4$ phenotype is accessible from multiple trajectories. Encouragingly, the trajectories predicted by *Heckman et al. (2013)* were found to pass through phenotypes of $C_3$–$C_4$ species, despite the fact that these species were not used to parameterise the evolutionary landscape. As different mechanisms generate increased abundance and cell-specificity for the majority $C_4$ enzymes in independent $C_4$ lineages (reviewed in *Langdale, 2011*; *Williams et al., 2012*), it is likely that mechanistic diversity underlies the multiple evolutionary pathways generating $C_4$ photosynthesis and may be a key factor in facilitating the convergent evolution of complex traits. This may benefit efforts to recapitulate the acquisition of $C_4$ photosynthesis through the genetic engineering of $C_3$ species (*Hibberd et al., 2008*), expanding the molecular toolbox available to establish $C_4$ traits in distinct phenotypic backgrounds.

## Methods

### Biological data from $C_4$ intermediates

Data from eighteen $C_3$, seventeen $C_4$, and thirty-seven $C_3$–$C_4$ species were consolidated from 43 studies that have examined the phenotypic characteristics of $C_3$–$C_4$ species since their discovery in

1974 (*Table 1*). Values for sixteen of the most widely studied $C_3$ characteristics were recorded for each intermediate species, as well as congeneric $C_3$ and $C_4$ relatives where available. The majority of data on enzyme abundance and the number and size of bundle sheath (BS) chloroplasts were obtained from studies employing the same methodology and were thus cross-comparable (e.g., *Goldstein et al., 1976*; *Ku et al., 1976*; *Ku and Edwards, 1978*; *Sayre et al., 1979*; *Holaday et al., 1981*; *Holaday and Black, 1981*; *Ku et al., 1983*; *Adams et al., 1986*; *Rajendrudu et al., 1986*; *Ku et al., 1991*; *Devi and Raghavendra, 1993*; *Bruhl and Perry, 1995*; *Casati et al., 1999*; *Keeley, 1999*). These cross-comparable quantitative data were partitioned into presence absence scores using two clustering techniques, the expectation-maximisation (EM) algorithm and hierarchical clustering (*Figure 1—figure supplement 3*). EM clustering was performed using a one-dimensional mixture model with two assigned components (e.g., presence and absence clusters), allowing for variable variance between the two components of the model, and variable population size between the two components. Hierarchical clustering was performed using a complete-linkage agglomerative approach, partitioning clusters by maximum distance according to a Euclidean distance metric. This approach identifies clusters with common variance, thus contrasting with the clusters of variable variance identifiable by EM.

For quantitative data not comparable with other studies (e.g., *Ku and Edwards, 1978*; *Rathnam and Chollet, 1978*; *Rathnam and Chollet, 1979*; *Holaday et al., 1984*; *Brown and Hattersley, 1989*; *Beebe and Evert, 1990*; *P'yankov et al., 1997*; *Gowik et al., 2011*), values obtained for intermediate species were scored as 1 or 0 if they were closer to the values for the respective $C_4$ or $C_3$ controls used in the original study. For qualitative abundance data from immunoblots (e.g., *Rosche et al., 1994*; *Voznesenskaya et al., 2007*; *Voznesenskaya, 2010*), relative band intensity was measured using ImageJ software (*Abramoff et al., 2004*) and abundance was scored as 1 or 0 if the band intensity value was closer to the $C_4$ or $C_3$ control respectively. For qualitative cell-specificity data from immunolocalisations (e.g. *Voznesenskaya et al., 2001*; *Ueno et al., 2003, 2006*; *Muhaidat et al., 2011*), a presence score was only assigned if the enzyme appeared completely absent from either mesophyll (M) or BS cells. We represent the phenotypic properties of each intermediate species as a string of $L = 16$ numbers (*Figure 1—source data 1*). We will refer to these strings as *phenotype strings* of L loci, with each locus describing data on the corresponding phenotypic trait. In a given locus, 0 denotes the absence of a $C_4$ trait, 1 denotes the presence of a $C_4$ trait, and 2 denotes missing data.

## Principal component analysis (PCA)

PCA was performed on five variables for $C_4$ cycle enzyme activity, with missing values estimated using the EM algorithm for PCA as described by *Roweis (1998)*.

## Model transition networks

The fundamental element underlying our analysis is a transition network $P$, consisting of a directed graph with $2^L = 65,536$ nodes, and the weight of the edge $P_{ij}$ denoting the probability of a transition occurring from node $i$ to node $j$. Each node corresponds to a possible phenotype: we labeled nodes with labels $l_i$ so that $l_i$ was the binary representation of the phenotype at node $i$, and $P_{ij}$ takes on the specific meaning of the probability of a transition from phenotype $l_i$ to phenotype $l_j$. We made several restrictions on the structure of $P$. We allowed only transitions that change a given phenotype at one locus, so $P_{ij} = 0$ if $H(l_i, l_j) \neq 1$, where $H(b_1, b_2)$ is the Hamming distance between bitstrings $b_1$ and $b_2$. Transitions that changed loci with value 1 to value 0 (steps back towards the $C_3$ state) were forbidden, so $P_{ij} = 0$ if $H(l_i, l_0) > H(l_j, l_0)$, where $l_0$ is the phenotypic string containing only zeroes. We assume that the possibility of events involving backwards steps, and multiple simultaneous trait acquisitions constitute second-order effects which will not strongly influence the inferred evolutionary dynamics.

## Evolutionary trajectories

Given the transition network $P$, we modelled the evolutionary trajectories that may give rise to $C_4$ photosynthesis through the picture of a discrete analogue to a Brownian bridge, that is as a stochastic process on $P$ with constrained start and end positions (*Revuz and Yor, 1999*). We enforced the start state of the process to be $l_{C_3} \equiv l_0 = 0...0$ (the phenotype string of all zeroes) and the end state, through the imposed structure of $P$, to be $l_{C_4} \equiv l_{2^L-1} = 1...1$ (the string of all ones). The dynamics of the process between these points consisted of $L$ steps, with a phenotypic trait being acquired at each step, and a step from node $i$ to node $j$ occurring with probability $P_{ij}$.

## Sampling intermediates

As many evolutionary trajectories may lead to the acquisition of the required phenotypic traits, we considered an ensemble of evolutionary trajectories for each transition network. Each member of this ensemble is started at $l_{C_3}$ and allowed to step across the network according to probabilities $P$.

To compare the dynamics of a given transition network to the properties of observed biological intermediates, we pictured this ensemble of trajectories as a modification of a hidden Markov model (HMM [*Rabiner, 1989*]). At each timestep in each individual trajectory, the process may with some probability emit a signal to the observer, with that signal being simply $l_i$, the label of the node at which the process currently resides. Over an ensemble of trajectories, a set of randomly emitted signals is thus built up (*Figure 1—figure supplement 4*).

We define a compatibility function between two strings as

$$C(s, t) = \prod_{i=1}^{L} c(s_i, t_i) \qquad (1)$$

$$c(s_i, t_i) = \begin{cases} 1 & \text{if } s_i = t_i \text{ or } s_i = 2 \text{ or } t_i = 2; \\ 0 & \text{otherwise.} \end{cases} \qquad (2)$$

$C(s,t)$ thus returns 1 if a signal comprising string $s$ could be responsible for observation $t$ once some of the loci within $s$ have been obscured: signal $s$ is compatible with observation $t$.

## Likelihood of observing biological data

We wish to compute the likelihood of observing biological data $B$ given a transition network $P$. Under our model, this likelihood is calculated by considering the compatibility of randomly emitted signals from processes supported by $P$ with the observed data $B$. We write

$$\mathcal{L}(P \mid B) = \prod_i \sum_{\text{chains } x} \sum_{\text{signals } s} \mathbb{P}_{\text{chain}}(x \mid P) \mathbb{P}_{\text{emission}}(s \mid x) C(s, B_i) \qquad (3)$$

Here, $\mathbb{P}_{\text{chain}}(x \mid P)$ is the probability of specific trajectory $x$ arising on network $P$, $\mathbb{P}_{\text{emission}}(s \mid x)$ is the probability that trajectory $x$ emits signal $s$, and $C(s, B_i)$ gives the compatibility of signal $s$ with intermediate state $B_i$. The term within the product operator thus describes the probability that evolutionary dynamics on network $P$ give rise to a signal that is compatible with species $B_i$, with the overall likelihood being the product of this probability over all observed species.

## Simulation

The uniform and random nature of signal emission means that $\mathbb{P}_{\text{emission}}(s \mid x)$ is a constant if signal $s$ can be emitted from trajectory $x$, and zero otherwise. Our simulation approach only produces signals which can be emitted from the trajectory under consideration, so $\mathbb{P}_{\text{emission}}(s \mid x)$ will always take the same constant value (which depends on the probability of signal emissions). As we will be considering ratios of network likelihoods and will not be concerned with absolute likelihoods we will ignore this term henceforth. For each network $P$ we simulate an ensemble of $N_{\text{chain}}$ trajectories and, for each node encountered throughout this ensemble, we record compatibilities with each of the biologically observed intermediates. We sum these compatibilities over the ensemble, obtaining $\sum_{\text{chains } x} \mathbb{P}_{\text{chain}}(x \mid P) C(s, B_i)$. A network that does not encounter any node compatible with a particular intermediate will thus be assigned zero likelihood; networks that encounter compatible nodes many times will be assigned high likelihoods.

For each transition network, we simulated $N_{\text{chain}} = 2 \times 10^4$ individual trajectories running from $C_3$ to $C_4$. This value was chosen after preliminary investigations to analyse the ability of trajectory ensembles to broadly sample available paths on networks.

## Bayesian MCMC over compatible networks

Given uninformative prior knowledge about the evolutionary dynamics leading to $C_4$ photosynthesis (specifically, our prior involves each possible transition from a given node being assigned equal probability), we aimed to build a posterior distribution over a suitable description of the evolutionary dynamics. We represented the dynamics supported on a network $P$ through a matrix $\pi$, where $\pi_{i,n}$

describes the probability that acquisition of trait $i$ occurs at the $n$th step in an evolutionary trajectory. The values of matrix $\pi_{i,n}$ were built up from sampling over the ensemble of trajectories simulated on $P$.

We used Bayesian MCMC to sample networks based on their associated likelihood values (**Wasserman, 2004**). At each iteration, we perturbed the transition probability of the current network $P$ a small amount (see below) to yield a new trial network $P'$. We calculated $\mathcal{L}(P'|B)$ and accepted $P$ as the new network if $\frac{\mathcal{L}(P'|B)}{\mathcal{L}(P|B)} > r$, where $r$ was taken from $\mathcal{U}(0,1)$. For practical reasons we implemented this scheme using log-likelihoods.

The perturbations we applied to transition probabilities are Normally distributed in logarithmic space: for each edge $w_{ij}$ we used $w'_{ij} = \exp\left(\ln w_{ij} + \mathcal{N}\left(0, \sigma^2\right)\right)$. To show that this scheme obeys detailed balance, consider two states $A$ and $A'$, for simplicity described by a one-dimensional scalar quantity. Consider the proposed move from $A$ when $\Delta$ is the result of the random draw. This proposal is $A \to A'$ if $A' = \exp(\ln A + \Delta) = Ae^{\Delta}$, implying that $A = A'e^{-\Delta}$. The probability of proposing move $A \to A'$ is thus $\mathcal{N}(x = \Delta \,|\, 0, \sigma^2)$, and the probability of proposing $A \to A'$ is $\mathcal{N}(x = -\Delta \,|\, 0, \sigma^2)$. By the symmetry of the Normal distribution, these two probabilities are equal.

We started each MCMC run with a randomly initialised transition matrix. We allowed $2 \times 10^4$ burn-in steps then sampled over a further $2 \times 10^5$ steps. The value $\sigma = 0.1$ was chosen for the perturbation kernel. These values were chosen through an initial investigation to analyse the convergence of MCMC runs under different parameterisations. We performed 40 MCMC runs for each experiment and confirmed that the resulting posterior distributions had converged and yielded consistent results.

### Summary dynamics matrices

We report the posterior distributions $\mathbb{P}(\pi_{i,n})$ inferred from sampling compatible networks as above. In the coarse-grained time representation, we use $\pi_{i,n'}^{CG} = \mathbb{P}\left(\sum_{n=1+4(n'-1)}^{4n'} \pi_{i,n}\right)$, summing over sets of ordinals of size 4.

We used the transition network $P$, rather than a more coarse-grained representation of the evolutionary dynamics (e.g., the summary matrices $\pi$), as the fundamental element within our simulations so as not to discard possible details that would be lost in a coarse-grained approach – for example, the presence of multiple distinct pathways, which may be averaged over in a summary matrix.

### Proofs of principle

To verify our approach, we constructed artificial data sets, consisting of sets of strings in which phenotypic traits were acquired in a single ordering. Specifically, $\pi_{i,n} = \delta_{i,n}$, so the first step always resulted in acquiring the first trait, and so on. To test the approach in a pleiotropic setting, where multiple traits were acquired simultaneously, we also constructed data sets where traits were acquired at only four timesteps, each corresponding to the simultaneous acquisition of four traits. We subjected these datasets to our inferential machinery with all data intact, and with 50% of data points occluded, to determine the sensitivity and robustness of our approach (**Figure 2A–B**). The approach accurately determines the ordering of events in both the bare and occluded cases and assigns very similar posterior probability distributions to the ordering of those traits acquired simultaneously.

### Comparing the evolution of multiple C$_4$ sub-types

To compare the pathways generating C$_4$ in monocots and eudicots, and in NADP-ME and NAD-ME sub-type lineages, we performed inference on two data sets: $B_1$ and $B_2$, each comprising phenotype measurements from one of the groups of interest. We reported the posteriors on the resulting summary dynamics $\mathbb{P}(\pi_{i,n})$ as before, and for the principal components analysis (PCA) we sampled $10^3$ summary dynamic matrices $\pi_{i,n}$ from the inferred posterior distribution during the Bayesian MCMC procedure, and performed PCA on these sampled matrices.

### Predictions

When a simulated chain encountered a phenotypic node compatible with a given biological intermediate, the values of traits corresponding to missing data in the biological data were recorded. These recorded values, sampled over the sampled set of networks, allowed us to place probabilities on the values of biologically unobserved traits inferred from the encounters of compatible dynamics with the corresponding phenotypic possibilities. For example, if 70% of paths on network $P$ pass through point 101 and

30% pass through point 001, we infer a 70% probability that the missing trait in biological intermediate 201 takes the value 1. Predictions were presented if the inferred probability of a '1' value was >75% (predicting a '1') or <25% (predicting a '0'). If one of these inequalities held and the limiting value fell outside one standard deviation of the inferred probability (i.e., for mean $\mu$ and standard deviation $\sigma$, $\mu > 0.75$ and $\mu - \sigma > 0.75$ [predicting a '1'] or $\mu > 0.25$ and $\mu + \sigma < 0.25$ [predicting a '0']), the prediction was presented as 'strict'.

### Acquisition ordering and evidence against a single pathway

We used a dynamic programming approach to explore whether a deterministic sequence of events, with a trait $T_n$ always being acquired at timestep $n$ ($\pi_{i,n} = \delta_{T_n,n}$), was compatible with the biological data. Performing an exhaustive search over sequences of single transitions that were compatible with the observed data, we identified several such sequences that accounted for all but one trait acquisition, but no single sequence exists that accounts for all the data.

### Contingent trait acquisition

To explore the possibility of multiple traits being acquired simultaneously, we tracked acquisition probabilities for later traits given that a certain trait was acquired first. This tracking was performed over all sampled compatible networks, building up 'contingent' acquisition tables $\gamma$ with the $i, j$ th element given by $\mathbb{P}(\pi_{j,2} \mid \pi_{i,1} = 1)$, $j \neq i$. If a pair of traits $i$ and $j$ were acquired simultaneously, we would expect $\gamma_{ij}$ and $\gamma_{ji}$ to both be higher than expected in the non-contingent case (as $j$ should always appear to be immediately acquired after $i$ and vice versa).

### Quantitative real-time PCR (qPCR)

RNA was extracted from mature leaves of six *Flaveria* species as part of the One Thousand Plants Consortium (www.onekp.com), using the hot acid phenol protocol as described by *Johnson et al. (2012)* (protocol no. 12). cDNA was synthesised from 0.5 µg RNA using Superscript II (Life Technologies, Glasgow, U.K.) following manufacturer's instructions. An oligo dT primer (Roche, Basel, Switzerland) was used to selectively transcribe polyadenylated transcripts. To each RNA sample, 1 fmol GUS transcript was added for use as an exogenous control or 'RNA spike', against which measured transcript abundance was normalised as described by *Smith et al. (2003)*.

qPCR was performed as described by *Bustin (2000)* using the DNA-binding marker SYBR Green (Sigma Aldrich, St. Louis, MO) according to manufacturer's instructions. Primers were designed using cDNA sequences for *Flaveria* species available at Genbank (http://www.ncbi.nlm.nih.gov/genbank) and synthesised by Life Technologies. Amplification was performed using a Rotor-Gene Q instrument (Qiagen, Hilden, Germany), using the following cycling parameters: 94°C for 2 min, followed by 40 cycles at 94°C for 20 s, 60°C for 30 s, 72°C for 30 s, followed by a 5 min incubation at 72°C. Relative transcript abundance was calculated as described by *Livak and Schmittgen (2001)*.

## Acknowledgements

We thank S Kelly, JA Langdale, H Griffiths, and N Jones for advice.

## Additional information

### Funding

| Funder | Author |
| --- | --- |
| Biotechnology and Biological Sciences Research Council | Ben P Williams, Iain G Johnston, Julian M Hibberd |
| International Rice Research Institute | Sarah Covshoff |

The funders had no role in study design, data collection and interpretation, or the decision to submit the work for publication.

### Author contributions

BPW, IGJ, Conception and design, Acquisition of data, Analysis and interpretation of data, Drafting or revising the article; SC, Acquisition of data, Analysis and interpretation of data, Drafting or revising the article; JMH, Conception and design, Analysis and interpretation of data, Drafting or revising the article

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
