## [Decision Letter]

Thank you for sending your work entitled “Phenotypic landscape inference reveals
multiple evolutionary paths to C_4_ photosynthesis” for consideration at
eLife. Your article has been favorably evaluated by a Senior editor, a Reviewing editor,
and 2 reviewers, one of whom, Patrick Warren, has agreed to reveal his identity.

The Reviewing editor and the two reviewers discussed their comments before we reached
this decision, and the Reviewing editor has assembled the following comments to help you
prepare a revised submission.

The manuscript by Williams et al. describes a systematic analysis of traits associated
with intermediate C_3_-C_4_ forms, and makes inferences on the likely
evolutionary pathways from C_3_ to C_4_ photosynthesis. It is largely
an in silico study, inferring the evolutionary dynamics within 16 traits that
distinguish C_3_ and C_4_ plants. The 18 lineages fall into 4 classes
within which the order of appearance of the traits is convergent. Furthermore, the model
predicts that there is a strong preference for the order of appearance of traits, a
feature the authors have validated by measuring some previously undetermined trait
values (a proxy being the abundance of transcripts by qPCR).

Although both reviewers noted that the work was predicated on the critical assumption
that present day intermediate forms are representative of the evolutionary pathways,
they found the work important from several angles. It suggests that the initial
phenotypic directions were unrelated to photosynthetic drivers, and only later co-opted
for a common end point. This is a valuable insight, and it seems to exemplify what could
be a generic mechanism to explain convergent evolution of complex traits. The work shows
just how far one can get with the systematic analysis of fragmented phenotype data, with
a cross-disciplinary approach, even so far as to make verifiable predictions for missing
trait data.

While agreeing that overall the paper is ambitious, has an evolutionary message, is
technically original and shows the predictive value of their approach, several
substantive concerns were raised that that should be addressed in the revision:

1) A strength of this work is that the conclusions emerge from an inference framework
that is much more convincing than when traditional (qualitative and argumentative)
approaches are used. Specifically, the methodology unveiled here is both quantitative
and “objective”. However, the authors need to be more explicit about the
level of “objectiveness” of their work. Indeed, they present a framework
where choices had to be made and the reader cannot know whether attempts with other
choices were less conclusive. To address this issue, the authors have to demonstrate
that their conclusions are robust to choices made within their modeling framework. Two
specific tests should be performed:

1A) First, use a structural change to your framework: instead of the 16 traits you have
selected, remove say 2 of these traits, and if possible include 2 others.

1B) Second, you treat quantitative traits using EM and represent these by a binary value
(presence vs absence). Sometimes the assignment will not be clear-cut; could you then
use the other assignment (which is nearly as justified)? An even simpler approach would
have been to simply put a threshold (common for all) bypassing any EM (you do this for
some of your traits). Did you first try that but not succeed? It is okay to be honest
about the potential weaknesses of the conclusions given that the data is limited.

2) To be clear to a broad audience, please be explicit about how you define a plant
‘lineage’ as used in your analysis. Your text describes that you have
analysed data from 18 lineages. Reference is made to ‘taxonomic lineage’
but in Table 1 there are 12 families and 22 genera, neither of which tallies with the
number 18. Also, please double check the species numbers: Table 1 appears to list 73
species. In the text you refer to 18 C_3_, 17 C_4_, and 37
C_3_-C_4_ intermediates, which totals 72 species. So, is there an
extra species listed in Table 1? Please check to make sure you (or the reviewers)
didn't miscount. A graphical representation of the phylogeny for the species used
in the analysis would be useful.

---

## [Author Response]

*1) A strength of this work is that the conclusions emerge from an inference
framework that is much more convincing than when traditional (qualitative and
argumentative) approaches are used. Specifically, the methodology unveiled here is
both quantitative and “objective”. However, the authors need to be more
explicit about the level of “objectiveness” of their work. Indeed, they
present a framework where choices had to be made and the reader cannot know whether
attempts with other choices were less conclusive. To address this issue, the authors
have to demonstrate that their conclusions are robust to choices made within their
modeling framework. Two specific tests should be performed*:

*1A) First, use a structural change to your framework: instead of the 16 traits
you have selected, remove say 2 of these traits, and if possible include 2
others*.

We performed three additional analyses. In the first two, we removed two randomly
selected independent pairs of traits (Figure 3—figure supplement 2). Neither the predicted timing of the
remaining 14 traits nor our main conclusions about C_4_ evolution were
affected. Thirdly, we repeated the analysis including data for two additional traits
associated with C_4_ evolution (Figure 3—figure supplement 3). Removing these traits from the analysis did
not alter the predicted order of trait acquisition compared with the initial analysis,
or the conclusions that we draw from these predictions.

*1B) Second, you treat quantitative traits using EM and represent these by a
binary value (presence vs absence). Sometimes the assignment will not be clear-cut;
could you then use the other assignment (which is nearly as justified)*?

To test this, we repeated the analysis with presence and absence scores assigned by
hierarchical clustering as opposed to EM. Hierarchical clustering generated alternative
binary values to EM for all the traits where assignment was not clear-cut. Despite this,
extremely similar predictions were generated and the conclusions we draw about
C_4_ evolution were the same. We include the results of this analysis in a
new supplement (Figure 3—figure supplement 1). We present both the data from hierarchical-clustered data on its own, as
well as a comparison with posterior probabilities obtained using data clustered by EM.
These results suggest that the data points whose assignment is not clear-cut do not
strongly affect our conclusions.

*An even simpler approach would have been to simply put a threshold (common for
all) bypassing any EM (you do this for some of your traits). Did you first try that
but not succeed? It is okay to be honest about the potential weaknesses of the
conclusions given that the data is limited*.

We note that both the EM algorithm and hierarchical clustering assign thresholds based
on the distribution of data available. We did not try assigning thresholds of our own
definition to these quantitative traits at any stage, but rather preferred to use
clustering by statistical methods such as EM so as to minimize bias in assigning
presence/absence scores. We only assigned our own thresholds to data for which
clustering by EM was not possible (i.e., when traits were measured qualitatively, or too
few data points were available).

We have integrated these new supplements associated with points 1A&B above into the
main article.

*2) To be clear to a broad audience, please be explicit about how you define a
plant ‘lineage’ as used in your analysis. Your text describes that you
have analysed data from 18 lineages. Reference is made to ‘taxonomic
lineage’ but in Table 1 there are 12 families and 22 genera, neither of which
tallies with the number 18. Also, please double check the species numbers: Table 1
appears to list 73 species. In the text you refer to 18
C*_*3*_*, 17
C*_*4*_
*and 37
C*_*3*_*-C*_*4*_
*intermediates, which totals 72 species. So, is there an extra species listed in
Table 1? Please check to make sure you (or the reviewers) didn't miscount. A
graphical representation of the phylogeny for the species used in the analysis would
be useful*.

We define the number of C_3_-C_4_ lineages by evolutionary independent
origins of C_3_-C_4_ intermediacy. Although our analysis included 15
genera possessing C_3_-C_4_ species, within two of these genera there
are multiple independent lineages of intermediates. For example, in Flaveria and Mollugo
there are three and two distinct clades of C_3_-C_4_ species
respectively. This totals 18 independent C_3_-C_4_ lineages. We have
annotated species in Table 1 as C_3_, C_4_, or
C_3_-C_4_ to provide further clarity. We included the following at
the beginning of the Results section to better clarify the definition of
intermediates:

“To parameterise the phenotypic landscape underlying photosynthetic phenotypes,
data was consolidated from 43 studies encompassing 18 C_3_, 18 C_4_,
and 37 C_3_-C_4_ intermediate species from 22 genera (Table 1). These
C_3_-C_4_ species are from 18 independent lineages likely
representing 18 distinct evolutionary origins of C_3_-C_4_
intermediacy (66) (Figure 1—figure supplement 2).”

Regarding the number of species, the 73 species listed in Table 1 is complete. The
numbers presented in the text were a count of these. A recount confirms that the
analysis included 18 C_3_, 18 C_4_, and 37 C_3_-C_4_
species. We are grateful for this anomaly being spotted.

We have included an angiosperm phylogeny with the distribution of independent
C_3_-C_4_ and C_4_ lineages annotated onto it (Figure 1—figure supplement 2).